# Scalable Gaussian Process via Hilbert-Schmidt Singular Value Decomposition

## Abstract

Gaussian process (GP) regression is widely used for its flexible mean predictions and inherent uncertainty quantification. However, its scalability is limited by cubic time complexity, $O(n^3)$, and quadratic space complexity, $O(n^2)$, making it infeasible for large-scale datasets. Although recent advances have introduced approximate methods with time complexity $O(nm^2)$, where $m \ll n$ is a tuning parameter, these methods each have their own bottlenecks, such as requiring a relatively large $m$ or involving expensive preprocessing steps. Moreover, for extremely large datasets with millions of samples, the space complexity $O(n^2)$ becomes another significant bottleneck. In this paper, we present a novel method based on the Hilbert-Schmidt singular value decomposition that obtains a low-rank decomposition "for free", reducing both time complexity to $O(nm^2)$ and space complexity to $O(nm)$, with no preprocessing overhead. We used simulated large-scale datasets to demonstrate the performance of our method compared to state-of-the-art approaches.

## 1 Introduction

Gaussian Process Regression (GPR) has become a cornerstone for nonparametric regression, largely due to its flexibility in modeling complex data and its inherent ability to provide uncertainty quantification (Cressie, 2015). GPR is widely used across numerous fields, including spatial and spatiotemporal modeling (Banerjee et al., 2003), epidemiology (Lawson et al., 2016), and machine learning (Rasmussen & Williams, 2006). Its ability to estimate both the conditional mean and covariance functions has made it indispensable for applications such as spatiotemporal data analysis, surrogate modeling for complex physical simulations, and Bayesian optimization.

However, a major challenge with GPR is its poor scalability. As size increases, GPR suffers from cubic time complexity, $O(n^3)$, due to the inversion of the kernel matrix, and quadratic space space complexity, $O(n^2)$, due to the need to store the kernel matrix, where $n$ is the number of data points. These computational constraints render GPR impractical for large-scale datasets, such as those found in forestry, geospatial applications, climate science, and single-cell RNA sequencing, where the sample size can reach millions or even tens of millions.

As a result, over the past two decades, significant progress has been made in scaling GPR for large datasets, with a primary focus on reducing computational complexity to $O(nm^2)$, where $m \ll n$ is a tuning parameter. These approaches can be broadly categorized into four groups: sparse approximations, low-rank approximations, probabilistic methods, and structured kernels.

Sparse approximations reduce the effective number of data points by using a subset of points. Examples of these methods include Sparse GPR (SGPR, Titsias, 2009, Nearest Neighbor Gaussian Process (NNGP, Finley et al., 2020), and approaches based on inducing points. Low-rank approximations aim to approximate the full kernel matrix using lower-dimensional representations. Common methods include the Nystrom approximation (Williams & Seeger, 2000) and random Fourier features (Rahimi & Recht, 2007). Additionally, methods like Deep Kernel Learning (DKL, Wilson et al., 2016) combine GPs with neural networks to learn effective representations from high-dimensional data, thereby reducing the dimensionality of the kernel. Probabilistic methods take a different route by optimizing the GP model through variational inference, allowing for efficient approximation of the posterior distribution. Examples include Variational Nearest Neighbor (VNN, Wu et al., 2022) and Sparse Variational Gaussian Process (SVGP, Hensman et al., 2015).

Variants like SVGP with Contour Integral Quadrature (SVGP-CIQ, Pleiss et al., 2020) further refine this approach by employing advanced numerical methods to enhance efficiency. These methods are often combined with Natural Gradient Descent (NGD, Salimbeni et al., 2018; Hensman et al., 2012) to further improve the computational efficiency of updating the variational parameters. Structured kernel approximations take advantage of specific structures in the covariance matrix to enable efficient computations. Methods like Structured Kernel Interpolation (SKI, Wilson & Nickisch, 2015) and KISS-GP (Wilson & Nickisch, 2015) belong to this category, exploiting grid-based or interpolation methods to approximate the kernel matrix more efficiently. Additionally, Krylov subspace methods, such as Lanczos Variance Estimates (LOVE, Pleiss et al., 2018), can approximate the kernel matrix with reduced complexity while maintaining high accuracy.

These various scalable GPR methods have significantly expanded the range of applications that GPs can handle in the context of big data. To streamline the use of these approaches, GPyTorch (Gardner et al., 2018) provides a powerful and efficient framework that integrates most of these methods into a single toolbox. By offering flexible options for scaling GPs, GPyTorch allows users to efficiently apply these techniques to large-scale datasets while maintaining robust predictive performance, making it an invaluable resource for researchers and practitioners alike.

Despite their advancements, these methods each have their own bottlenecks. For instance, some approaches require a relatively large $m$, reducing their efficiency, while others involve complex preprocessing steps or have tuning parameters that are difficult to optimize. Additionally, for extremely large datasets, with millions or tens of millions of samples, the space complexity—which has been relatively less addressed by many of these methods—becomes a significant bottleneck. Moreover, methods like DKL and some variants of SVGP often require GPUs to manage the computational load, which may not be readily available to all practitioners. This highlights the need for a method that is both time- and space-efficient, requires minimal tuning or preprocessing, and does not depend on GPU resources—though it should still be able to benefit from GPU acceleration when available.

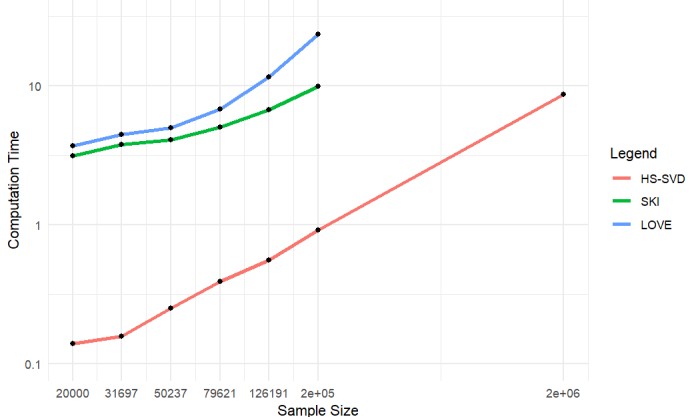

Figure 1: Runtime (second) of our proposed HS-SVD and two SOTA methods: SKI and LOVE.

In this paper, we propose a scalable Gaussian Process framework based on the Hilbert-Schmidt Singular Value Decomposition (HS-SVD), achieving a time complexity of $O(nm^2)$ and a space complexity of $O(nm)$ while maintaining strong predictive performance. Our method leverages a specifically designed family of kernels, including the compact Matérn kernel, for which we prove its smoothness. There is only one easy-to-tune parameter, $m$, an integer, and requires no preprocessing. Our method does not require a GPU, making it accessible to a wider range of users. However, when available, GPU usage can further accelerate the computations, providing an additional speed advantage without being a necessity. Figure 1 shows that on simulated data (see Section 4 for more details), our proposed method, without using any GPUs, is more efficient than two state-of-the-art (SOTA) methods, SKI and LOVE, using GPUs. We further validate the superior performance of our approach on four simulated large-scale datasets, showing its advantage in terms of prediction MSE, runtime, and memory usage, even compared to methods that rely on GPUs.

## 2 PRELIMINARIES

In this section, we review some of the key concepts and results in GPR and Hilbert-Schmidt Singular Value Decomposition (HS-SVD), upon which our later methods will hinge.

### 2.1 GAUSSIAN PROCESS REGRESSION

GPR is a nonparametric regression technique, particularly valued for its interpretability and strong theoretical foundations (Stein, 2012). The goal of GPR is to predict the values of a function based on noisy observations. We begin with the definition of a GP.

**Definition 2.1 (GP)** *$f$ is said to follow a GP over domain $\Omega$ with mean function $\mu : \Omega \to \mathbb{R}$ and covariance or kernel function $K : \Omega \times \Omega \to \mathbb{R}$, denoted by $f \sim \mathrm{GP}(\mu, K)$, if for any $x_1, ..., x_n \in \Omega$,*

$$[f(x_1), \cdots, f(x_n)]^\top \sim N(v, \Sigma), \text{ where } v = [\mu(x_1), \cdots, \mu(x_n)]^\top, \Sigma_{ij} = K(x_i, x_j).$$

GPR takes a Bayesian approach by estimating the posterior distribution of the function given a set of observations $X = (x_1, .., x_n), Y = (y_1, ..., y_n)$.

**Definition 2.2 (Posterior of GP)** *Let $f \sim \mathrm{GP}(\mu, K)$. Then, given observations $X, Y$, $f(Z)$, where $Z$ is a new set of points, is estimated by its posterior $f(Z)|(X, Y)$, given by the following form:*

$$f(Z)|(X, Y) \sim N\left(\mu(Z) + K_{ZX}K_{XX}^{-1}(Y - \mu(X)), K_{ZZ} - K_{ZX}K_{XX}^{-1}K_{XZ}\right),$$

*where*

$$K_{XZ} = \begin{bmatrix} K(z_1, x_1) & K(z_1, x_2) & ... & K(z_1, x_n) \\ K(z_2, x_1) & K(z_2, x_2) & ... & K(z_2, x_n) \\ \vdots & & & \\ K(z_m, x_1) & K(z_m, x_2) & ... & K(z_m, x_n) \end{bmatrix}.$$

In GPR, the primary computational challenge lies in calculating $K_{XX}^{-1}$, which requires $O(n^3)$ operations. Moreover, storing $K_{XX}$ requires a space complexity of $O(n^2)$, as the entire kernel matrix must be retained in memory. Another issue arises from numerical stability. As the data become denser, or as the minimal separation between data points approaches zero, the matrix tends to become ill-conditioned. This results in numerical instability during inversion, even when the time complexity may be theoretically manageable. Hence, the combination of computational cost and potential instability creates significant challenges for applying GPR to large datasets. These challenges have motivated a range of approximation techniques, such as low-rank approximations and the use of inducing points, each with its own trade-offs in terms of accuracy, computational cost, and stability.

### 2.2 HILBERT-SCHMIDT SINGULAR VALUE DECOMPOSITION

We first define the Mercer decomposition of kernels.

**Lemma 2.3 (Mercer decomposition)** *Let $K$ be a continuous positive definite kernel on $\Omega$. $K(x, x')$ can be expressed by the following form:*

$$K(x, x') = \sum_{l=1}^{\infty} \lambda_l \phi_l(x) \phi_l(x'),$$

*where $\lambda_1 \geq \lambda_2 \geq ... \to 0$ and $\phi_l$'s are orthonormal.*

This decomposition is also known as the Hilbert-Schmidt series, where $\phi_l$'s are eigenfunctions of the associated integral operator. We then use Mercer's theorem to define the Hilbert-Schmidt Singular Value Decomposition (HS-SVD) for a kernel matrix.

**Definition 2.4 (HS-SVD)** *Let $K(x, x') = \sum_{l=1}^{\infty} \lambda_l \phi_l(x) \phi_l(x')$ be the Mercer decomposition of kernel $K$, and consider the kernel matrix $\Sigma = K_{XX}$. Then the HS-SVD of $\Sigma$ is given by*

$$\Sigma = \Phi \Lambda \Phi^{\top},$$

*where*

$$\Phi^{\top} = \begin{bmatrix} \phi_1(x_1) & \dots & \phi_1(x_n) \\ \phi_2(x_1) & \dots & \phi_2(x_n) \\ & \vdots & \\ \phi_\infty(x_1) & \dots & \phi_\infty(x_n) \end{bmatrix}, \Lambda = \begin{bmatrix} \lambda_1 & & & \\ & \lambda_2 & & \\ & & \ddots & \\ & & & \lambda_\infty \end{bmatrix}.$$

*We note that the notation of infinite matrices is formal, while for computation we use the truncated form whose depth may be chosen based on the numerical limits of one's data type. The truncation strikes a balance between computational efficiency and the accuracy of the low-rank approximation. In this paper, we will use the following approximation*

$$\Sigma \approx \Sigma_m := \Phi_m \Lambda_m \Phi_m^{\top} = \begin{bmatrix} \phi_1(x_1) & \dots & \phi_m(x_1) \\ \phi_1(x_2) & \dots & \phi_m(x_2) \\ & \vdots & \\ \phi_1(x_n) & \dots & \phi_m(x_n) \end{bmatrix} \begin{bmatrix} \lambda_1 & & & \\ & \lambda_2 & & \\ & & \ddots & \\ & & & \lambda_m \end{bmatrix} \begin{bmatrix} \phi_1(x_1) & \dots & \phi_1(x_n) \\ \phi_2(x_1) & \dots & \phi_2(x_n) \\ & \vdots & \\ \phi_m(x_1) & \dots & \phi_m(x_n) \end{bmatrix}.$$

*$\Sigma_m$ is a rank-$m$ approximation to $\Sigma$, which is the foundation of our proposed methods in this paper.*

## 3 METHOD

As discussed in Section 2.1, the primary computational bottlenecks in GPR arise from the inversion of the kernel matrix $K_{XX}$, which incurs a time complexity of $O(n^3)$ and a space complexity of $O(n^2)$. Current scalable methods typically approach these problems using two common strategies:

Low-rank approximation: These methods approximate $K_{XX}$ with a low-rank representation to make computations more tractable. However, the cost of computing this low-rank representation is often significant, and in practice, it may introduce numerical errors that compromise the model's performance.

Sparse approximation: This approach reduces the size of $K_{XX}$ by using a subset of inducing points to approximate the dataset. The challenge with this method lies in choosing the number of inducing points: too few points compromise accuracy, while too many reintroduce the computational burden of a large matrix.

In this section, we propose a novel framework that addresses both size and stability challenges simultaneously. Our method leverages a low-rank decomposition of the kernel matrix without incurring the high computational cost of explicitly calculating this decomposition. We achieve these goals by using the HS-SVD introduced in Section 2.2, which is based on the Mercer decomposition of the kernel. This allows us to bypass the direct computation of the low-rank form while still retaining the advantages of the decomposition.

### 3.1 COMPACT MATÉRN KERNELS

As discussed in Section 2.2, HS-SVD provides an efficient way to approximate the kernel matrix $K_{XX}$ by exploiting its spectral properties. Specifically, we have $K_{XX} \approx K_m = \Phi_m^{\top} \Lambda_m \Phi_m$. The choice of $m$ is crucial for ensuring both accuracy and computational efficiency. The decay rate of the eigenvalues $\lambda_i$ typically governs how quickly the truncation error diminishes. For well-behaved kernels, the decay rate is at least polynomial, allowing for rapid convergence of the truncated approximation to the full kernel.

This method works for any kernel, provided that its Mercer decomposition is known or its eigenfunctions $\phi_i$ and eigenvalues $\lambda_i$ can be computed. However, for several widely used kernels, such as the Radial Basis Function (RBF), Matérn, and exponential kernels, the exact Mercer decomposition is not straightforward or explicitly available. To address this, we propose using the compact Matérn kernel, which was initially defined for the 1-Dimensional interval: $\Omega = [0, 1]$ in Cavoretto et al.

(2015), to study the radial basis function. We then extend this definition to arbitrary dimensions, providing a practical and computationally efficient alternative.

**Definition 3.1 (1-D compact Matérn, Cavoretto et al., 2015)** *The 1-Dimensional compact Matérn kernel, denoted by $K_{\rho,\alpha,\beta}^c : [0,1] \times [0,1] \to \mathbb{R}$, parametrized by $\alpha, \rho > 0$ and $\beta \in \mathbb{Z}_{>0}$, is defined as*

$$K_{\rho,\alpha,\beta}^c(x,x') = \rho \sum_{l=1}^{\infty} \lambda_l \phi_l(x) \phi_l(x') = \rho \sum_{l=1}^{\infty} 2(\alpha^2 + l^2\pi^2)^{-\beta} \sin(l\pi x) \sin(l\pi x'),$$

*where $\lambda_l = (\alpha^2 + l^2\pi^2)^{-\beta}, \phi_l(x) = \sqrt{2}\sin(l\pi x)$.*

In this kernel, $\rho$ is often called the spatial variance or partial sill that controls the point-wise variance, $\alpha$ is often referred to as the lengthscale, tension, or decay parameter that measures the spatial dependency, and $\beta$ is called the smoothness parameter.

While the 1-D compact Matérn kernel serves as a valuable kernel, many practical applications require an extension to higher dimensions, especially 2-D for geospatial applications. We now generalize it to arbitrary dimension $r$:

**Definition 3.2 (Compact Matérn)** *The compact Matérn kernel, denoted by $K_{\rho,\alpha,\beta}^c : [0,1]^r \times [0,1]^r \to \mathbb{R}$, parametrized by $\alpha, \rho > 0$ and $\beta \in \mathbb{Z}_{>0}$, is defined as*

$$K_{\rho,\alpha,\beta}^c(x,x') = \rho \sum_{l_1,..,l_r=1}^{\infty} \lambda_{l_1,...,l_r} \phi_{l_1,..,l_r}(x) \phi_{l_1,..,l_r}(x'),$$

*where $\lambda_{l_1,...,l_r} = \left(\alpha^2 + \pi^2(\sum_{q=1}^r l_q^2)\right)^{-\beta}, \phi_{l_1,..,l_r}(x) = \sqrt{2}^r \prod_{q=1}^r \sin(l_q \pi x_q)$.*

Next, we prove the smoothness of the compact Matérn kernel, which explains why $\beta$ is called the smoothness parameter.

**Theorem 3.3** *The compact Matérn kernel is $\beta - r - 1$ times differentiable.*

This smoothness result provides flexibility in controlling the regularity of the function sampled from the GP. By tuning the parameter $\beta$, practitioners can adjust the differentiability of the resulting process, making the compact Matérn kernel an attractive choice for applications requiring different levels of smoothness.

This flexibility in smoothness is also one of the key benefits of the standard Matérn kernel (Stein, 2012). This connection offers a fair basis for comparison with other methods. For example, we can directly compare our method with a $C^1$ compact Matérn and another method, such as NNGP, using a standard $C^1$ Matérn kernel. This ensures that any differences in performance can be attributed to the underlying method, rather than differences in the choice of kernel smoothness.

Although the compact Matérn kernel may initially appear arbitrary, its definition is based on a theoretical connection to a differential operator. Specifically, we derive this kernel as the Green's function of the modified Helmholtz operator, which further demonstrate the relationship between compact Matérn and standard Matérn.

**Definition 3.4 (Modified Helmholtz operator)** *Let $\Delta$ denote the Laplacian. Then define the following modified Helmholtz operator $L_{\beta,\alpha} : L_2(\Omega) \to L_2(\Omega)$ as*

$$L_{\beta,\alpha} = (-\Delta + \alpha^2 \,\mathrm{I})^\beta,$$

*where $\Delta$ is the Laplacian.*

**Proposition 3.5 (Cavoretto et al., 2015)** *Let $K(x,x') : \Omega \times \Omega \to \mathbb{R}$ be the Green's function of $L_{\beta,\alpha}$.*

- *When $\Omega = \mathbb{R}^r$, then $K(x, x')$ is the standard Matérn Kernel over $\mathbb{R}^r$:*

$$K(x, x') = (\alpha \|x - x'\|)^{\beta - r/2} \, K_{\beta - r/2}(\alpha \|x - x'\|), \ \beta > \frac{r}{2},$$

  *where $K_{\beta - r/2}$ is the modified Bessel function of the second kind with degree $\beta - r/2$.*

- *When $\Omega = [0, 1]^r$ with zero boundary condition, then $K(x, x')$ is the compact Matérn kernel in Definition 3.2.*

In summary, the compact Matérn kernel and the standard Matérn kernel are fundamentally the same at the differential operator level, with the compact Matérn defined on a compact domain $[0, 1]^r$, while the standard Matérn operates over the entire Euclidean space $\mathbb{R}^r$. Both kernels arise from formally identical differential operators; however, the boundary conditions differ based on the domain. It is also worth noting that the interval $[0, 1]$ is used for simplicity of presentation, and the domain can be replaced with any closed interval or bounded region without loss of generality.

## 3.2 FAST PREDICTION

As discussed in Section 2.1, the posterior mean for prediction, also known as the best linear unbiased prediction (BLUP), is given by $\mathbb{E}(f(Z)|X, Y) = \mu(Z) + K_{ZX} K_{XX}^{-1}(Y - \mu(X))$. For simplicity, we assume $\mu \equiv 0$, as the prior mean can be handled separately. Then the BLUP becomes $K_{ZX} K_{XX}^{-1} Y$. The main computational burden GPR arises from the inversion $K_{XX}^{-1}$. Therefore, improving the computational efficiency largely depends on the stable and quick inverse of the kernel matrix. To account for noise or measurement error, which is a reasonable assumption in many real-world settings, a nugget term, $\sigma^2$, is often added to the diagonal of the kernel matrix. This term represents independent noise added to each observation, improving the model's fit to noisy data. Additionally, the nugget helps with numerical stability, particularly when the data is dense or there are small separations between points, by ensuring that the kernel matrix remains well-conditioned during inversion. Combining the HS-SVD with the Sherman–Morrison–Woodbury formula (Pozrikidis, 2014), we can approximate the inverse as:

$$K_{XX}^{-1} = (\sigma^2 \, \mathrm{I}_n + \Phi \Lambda \Phi^\top)^{-1} \approx (\sigma^2 \, \mathrm{I}_n + \Phi_m \Lambda_m \Phi_m^\top)^{-1} = \frac{1}{\sigma^2} \left( \mathrm{I}_n - \Phi_m (\sigma^2 \Lambda_m^{-1} + \Phi_m^\top \Phi_m)^{-1} \Phi_m^\top \right).$$

Since the inversion only involves $\sigma^2 \Lambda_m^{-1} + \Phi_m^\top \Phi_m$, an $m \times m$ matrix, the computational complexity is reduced to $O(m^3)$, where $m \ll n$, providing significant improvements over direct inverting the $n \times n$ kernel matrix.

## 3.3 FAST LIKELIHOOD EVALUATION

In the previous section, we focused on making prediction using BLUP, which requires the kernel matrix $K$, parameterized by kernel parameter $\theta_K$ and nuggets $\sigma^2$. These parameters are typically unknown and must be estimated. The most common approach for estimating $\theta$ is the Maximum Likelihood Estimation (MLE). In this section, we address how to efficiently compute the log-likelihood using the HS-SVD framework.

Up to an additive constant, the log likelihood function of a GP is

$$\ell(\theta) = -\frac{1}{2} Y^\top K(\theta)^{-1} Y - \frac{1}{2} \log |K(\theta)|. \tag{1}$$

Observe that in compact Matérn, only the eigenvalues $\lambda_j(\theta)$ depend on the parameters $\theta$, while the eigenfunctions $\phi_j(x)$ do not depend on $\theta$. This assumption holds for many common kernels, including RBF, standard Matérn, and exponential kernels. Under this assumption, the matrices $\Phi_m$ do not need to be recomputed at each iteration of the optimization, allowing for efficient likelihood evaluation.

Using the HS-SVD, we can express the first term in Equation (1) efficiently:

$$Y^\top K(\theta)^{-1} Y = \frac{1}{\sigma^2} Y^\top \left[ \mathrm{I}_n - \Phi_m (\sigma^2 \Lambda_m(\theta)^{-1} + \Phi_m^\top \Phi_m)^{-1} \Phi_m^\top \right] Y.$$

Again using HS-SVD, together with the Sylvester's determinant theorem (Pozrikidis, 2014), the log determinant in the second term of Equation (1) can be computed as

$$\log \det(\sigma^2 \, \mathrm{I}_n + \Phi_m \Lambda_m(\theta) \Phi_m^\top) = \log \left[ (\sigma^2)^n \det(\mathrm{I}_n + \frac{1}{\sigma^2} \Phi_m \Lambda_m(\theta) \Phi_m^\top) \right]$$

$$= n \log(\sigma^2) + \log \det \left( \mathrm{I}_m + \frac{1}{\sigma^2} \Phi_m^\top \Phi_m \Lambda_m(\theta) \right).$$

Combining these two components, we arrive at the following approximation to the log-likelihood:

$$\frac{1}{\sigma^2} Y^\top \left[ \mathrm{I}_n - \Phi_m(\sigma^2 \Lambda_m(\theta)^{-1} + \Phi_m^\top \Phi_m)^{-1} \Phi_m^\top \right] Y + n \log(\sigma^2) + \log \det \left( \mathrm{I}_m + \frac{1}{\sigma^2} \Phi_m^\top \Phi_m \Lambda_m(\theta) \right).$$

The reason for the speedup is due to the reduction in the size of the matrix computations. Without the HS-SVD approximation, evaluating $K(\theta)^{-1}$ and $\log |K(\theta)|$ involves operating on large $n \times n$ matrices, which is both computationally and memory intensive. By using the HS-SVD, these operations are reduced to much smaller $m \times m$ matrices, where $m \ll n$.

Furthermore, the truncation of eigenvalues in the HS-SVD approximation not only speeds up computation but also stabilizes it. The discarded eigenvalues often approach zero and contribute little to the overall matrix, but can make the original kernel matrix ill-conditioned. Truncating these small eigenvalues removes the "singular" parts of the matrix, improving the stability of the inverse and ensuring that the computations remain stable as $n$ increases. In this way, our approach not only accelerates the computations but also enhances their numerical stability.

### 3.4 ALGORITHM: THE HS-SVD-BASED SCALABLE GP

In this section, we present the complete scalable GP algorithm based on HS-SVD, as discussed in previous sections. We keep the algorithm general, for kernels with a known Mercer decomposition whose eigenfunctions are independent of the kernel parameters, including compact Matérn as a concrete family of kernels of this kind. This method takes in data $X, Y$ and a set of new points $X_{new}$, and outputs predicted outcomes $Y_{new}$ at these new points. In addition to parameter estimation by MLE, this method can easily be extended to include posterior samples for uncertainty quantification, since the key challenge, efficiently calculating $K^{-1}$, has already been addressed.

**Algorithm**: HS-SVD-based scalable GP regression
**Require:** $m, \theta = (\theta_K, \sigma^2), X = (x_1, ..., x_n), Y = (y_1, ..., y_n), X_{new} = (x'_1, ..., x'_p)$
**Ensure:** $m \leq n$
    $\Phi_{il} \leftarrow \phi_l(x_i)$ for $x_1, ..., x_n$ and $\phi_1, ..., \phi_m$
    Memory $\leftarrow \Phi^\top Y$
    Memory $\leftarrow \Phi^\top \Phi$
    $\ell(\theta) := \frac{1}{\sigma^2} Y^\top \left[ \mathrm{I}_n - \Phi(\sigma^2 \Lambda(\theta)^{-1} + \Phi^\top \Phi)^{-1} \Phi^\top \right] Y + n \log(\sigma^2) + \log \det(\mathrm{I}_m + \frac{1}{\sigma^2} \Phi^\top \Phi \Lambda(\theta))$
    $\hat{\theta} = (\hat{\theta}_K, \hat{\sigma^2}) = \mathrm{argmin}_\theta \, -2\ell(\theta)$
    $\hat{\Lambda} \leftarrow (\lambda_1(\hat{\theta}), ..., \lambda_m(\hat{\theta}))$
    $G \leftarrow (\hat{\sigma}^2 \hat{\Lambda}^{-1} + \Phi^\top \Phi)^{-1}$
    $H \leftarrow \frac{1}{\sigma^2} \left( Y - \Phi G \Phi^\top Y \right)$
    $\Phi'_{il} \leftarrow \phi_l(x'_i)$ for $x'_1, ..., x'_p$ and $\phi_1, ..., \phi_m$
    $Y_{new} \leftarrow \Phi' \hat{\Lambda} \Phi^\top H$
    Output: $Y_{new}$

It is important to note that the large matrices in this algorithm should be stored in their low-rank decompositions rather than their full-size forms, to improve computational efficiency and memory savings. Any quantity that does not depend on $\theta$, including $Y^\top Y, \Phi^\top Y$ and $\Phi^\top \Phi$, should only be calculated once and then stored in memory. Furthermore, when evaluating matrix multiplications, we always evaluate from right to left in order to use matrix-vector multiplications rather than matrix-matrix multiplications.

## 4 SIMULATIONS

In this section, we conduct numerical experiments to evaluate the effectiveness of our method compared to nine representative SOTA methods. We simulate large-scale datasets with $n = 100,000$ from highly nonlinear functions, and compare the methods based on prediction mean square error (MSE), run time, and RAM and VRAM usage.

The methods in our benchmark are: our proposed HS-SVD, Nearest Neighbor Gaussian Process (NNGP), Stochastic Variational Gaussian Process (SVGP), Stochastic Variational Gaussian Process with Contour Integral Quadrature (SVGP-CIQ), Variational Nearest Neighbor (VNN), Natural Gradient Descent (NGD), Deep Kernel Learning (DKL), Sparse Gaussian Process Regression (SPGR), Structured Kernel Interpolation (SKI), and the Lanczos Variance Estimates (LOVE). The HS-SVD method is implemented by our own R code, the NNGP implementation is from the spNNGP R package, and the remaining implementations are from the GPyTorch Python library. In the HS-SVD simulations, a single CPU core was used. In the NNGP method, 16 CPU threads were used. The remaining methods all used GPU(s).

In Simulation 1-4, we generate $\sim 100,000$ samples, and randomly select $80,000$ for training and $\sim 20,000$ for testing, with 10 replicates in each setup. MSE and time are reported as mean (standard deviation) in all tables, while RAM and VRAM usage are based on peak usage during a single run, because memory usage depends heavily on external factors, such as system state, fragmentation of memory, and caching. RAM usages should be judged in a relative manner, and VRAM usage primarily indicates the maximum batch size that allows for convergence. Simulation 5 follows the same setup as Simulation 1, except that the sample size varies from $20,000$ to $2,000,000$ to produce the results shown in Figure 1 in Section 1. For a fair comparison, we ensure that the kernel for each model has the same smoothness. For the HS-SVD method, we chose $\beta = 3$ in the 1-D case, and $\beta = 4$ in the 2-D case. Correspondingly, for the remaining methods, we chose Matérn with $\nu = 3/2$.

**Simulation 1**
In our first simulation, we begin with the 1-Dimensional case using a complex function, $y = \sin(300(x - 0.5)^2)$ (Figure 2 left). The $100,000$ $x$ values are evenly spaced on $[0.2, 0.8]$. Gaussian noise with $\sigma^2 = 0.3$ is added to $y$. Table 1 shows that HS-SVD with $m = 50$ performs the best among these methods in terms of MSE, time, and memory.

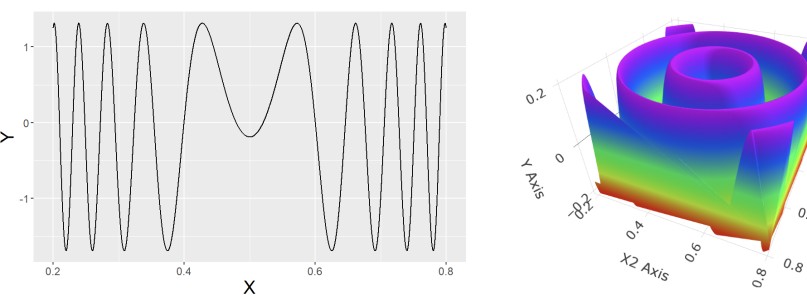

Figure 2: Data generating functions for Simulations 1, 2 and 5 (left) and Simulation 3 and 4 (right)

**Simulation 2**
In the second simulation, we repeat the setup of Simulation 1 with one major change: the data are no longer uniform across the domain. Instead, half of the data are uniform across the domain $[0.2, 0.8]$, and another half of the data are concentrated uniformly in the small region $[0.25, 0.251]$. This setup is challenging as the kernel matrix becomes increasingly ill-conditioned as the data concentrates around a single point $(0.25)$. Table 2 shows that the MSE of NNGP and SVGP increased significantly due to the ill-conditioned kernel matrix, while other methods still perform well.

**Simulation 3**
In the third simulation, we extend the evaluation to 2-Dimensional inputs. The true function is given by $y = 0.2\sin(100||x - [0.5, 0.5]^\top||_2^2)$ (Figure 2 right). The design points are evenly distributed on a $317 \times 317$ grid on the square $[0.2, 0.8]^2$ (with $100,489$ total samples), then randomly split into $80,000$ for training and $20,489$ for testing. Gaussian noise with $\sigma^2 = 0.1$ is added to $y$.

Table 1: Results from Simulation 1, with evenly spaced 1-D inputs.

| Method | MSE | (SD) | Time (seconds) | (SD) | RAM (MB) | VRAM (MB) |
|---|---|---|---|---|---|---|
| HS-SVD-50 | **0.2991** | (<0.001) | **0.42** | (0.01) | **135.2** | **0** |
| NNGP | 0.3158 | (<0.001) | 104.49 | (1.01) | 142.7 | **0** |
| SVGP | 0.3015 | (0.001) | 164.51 | (23.35) | 316.78 | 17.25 |
| SVGP-CIQ | 0.3011 | (<0.001) | 15.77 | (0.11) | 487.82 | 40.25 |
| VNN | 0.2998 | (<0.001) | 367.02 | (33.57) | 1177.16 | 2731.36 |
| NGD | 0.3045 | (<0.001) | 166.24 | (8.33) | 347.58 | 196.16 |
| DKL | 0.3015 | (0.007) | 29.90 | (0.45) | 724.95 | 515.43 |
| SGPR | 0.3020 | (<0.001) | 22.48 | (1.26) | 411.61 | 99.64 |
| SKI | 0.2996 | (<0.001) | 6.61 | (0.57) | 817.19 | 35.02 |
| LOVE | 0.2992 | (<0.001) | 38.10 | (0.42) | 698.21 | 515.12 |

Table 2: Results from Simulation 2, with half evenly spaced and half dense 1-D inputs.

| Method | MSE | (SD) | Time (seconds) | (SD) | RAM (MB) | VRAM (MB) |
|---|---|---|---|---|---|---|
| HS-SVD-50 | **0.3046** | (<0.001) | **0.426** | (0.015) | **130.9** | **0** |
| NNGP | 0.3546 | (0.030) | 104.331 | (1.59) | **132.3** | **0** |
| SVGP | 0.3346 | (0.003) | 158.99 | (3.44) | 314.90 | 17.66 |
| SVGP-CIQ | 0.3091 | (0.001) | 42.30 | (2.99) | 480.98 | 40.25 |
| VNN | 0.3057 | (<0.001) | 356.59 | (32.91) | 1161.08 | 2731.97 |
| NGD | 0.3099 | (0.001) | 173.27 | (5.57) | 348.04 | 196.16 |
| DKL | 0.3048 | (<0.001) | 35.15 | (0.99) | 704.00 | 532.98 |
| SGPR | 0.3054 | (<0.001) | 21.70 | (1.27) | 412.84 | 99.63 |
| SKI | 0.3050 | (<0.001) | 6.20 | (0.16) | 673.13 | 35.02 |
| LOVE | **0.3046** | (<0.001) | 42.66 | (2.37) | 681.31 | 514.16 |

Table 3: Results from Simulation 3, with evenly spaced 2-D inputs.

| Method | MSE | (SD) | Time (seconds) | (SD) | RAM (MB) | VRAM (MB) |
|---|---|---|---|---|---|---|
| HS-SVD-169 | **0.1010** | (<0.001) | **4.14** | (0.13) | 337.1 | **0** |
| NNGP | 0.1045 | (<0.001) | 113.48 | (1.05) | **148.9** | **0** |
| SVGP | 0.1187 | (<0.001) | 206.71 | (2.79) | 779.40 | 98.018 |
| SVGP-CIQ | 0.1079 | (<0.001) | 26.06 | (0.33) | 480.61 | 17.97 |
| VNN | 0.1012 | (<0.001) | 24.89 | (1.38) | 975.70 | 1267.19 |
| NGD | 0.1013 | (<0.001) | 61.47 | (0.26) | 362.33 | 26.33 |
| DKL | 0.1148 | (0.003) | 21.07 | (1.88) | 679.79 | 516.29 |
| SGPR | 0.1029 | (<0.001) | 42.78 | (3.41) | 413.25 | 100.03 |
| SKI | 0.1209 | (<0.001) | 157.68 | (12.16) | 3903.85 | 32.20 |
| LOVE | 0.1169 | (<0.001) | 18.05 | (0.29) | 661.31 | 515.32 |

Table 3 shows a similar trend to the 1d case: the HS-SVD case achieves an MSE that is comparable to the best performing methods while maintaining an excellent runtime. However, in this 2-D scenario, HS-SVD requires more RAM than NNGP, primarily due to the need to store a larger number of eigenfunctions, 169 in this 2-D case versus 50 in the 1-D case. Despite the higher memory usage, the significant reduction in runtime justifies this trade-off. Moreover, methods that rely on interpolation grids, such as SKI, scaled poorly with input dimension $r$, especially when the true function is complex, as the number of grid points grows exponentially with $r$. On the other hand, the methods that scale the most efficiently with $r$ are those that either incorporate dimension reduction (DR), such as in DKL and LOVE, or use variational inference, such as VNN and SVGP-CIQ. Importantly, these methods are fully compatible with our HS-SVD framework. By integrating the strength of DR and variational inference methods with HS-SVD's low-rank decomposition, we could achieve the best of both worlds—enhancing stability while maintaining fast runtime and low memory costs.

**Simulation 4**

In the fourth simulation, we repeat the setup of Simulation 3 with the addition of an extremely dense region. Half of the design points are evenly distributed on a $234 \times 234$ grid on the square $[0.2, 0.8]^2$. Another half of the design points are on a $234 \times 234$ grid is placed in the small region $[0.25, 0.251]^2$, leading to $109,512$ total samples. The dataset is then randomly split into a $80,000$ training set and a $29,512$ test set. Table 4 shows a similar trend as in Simulation 3, where with HS-SVD showing the best MSE and runtime, while using more RAM than NNGP due to the larger number of required eigenfunctions.

Table 4: Results from Simulation 4, with half evenly spaced and half dense 2-D inputs.

| Method | MSE | (SD) | Time (seconds) | (SD) | RAM (MB) | VRAM (MB) |
|---|---|---|---|---|---|---|
| HS-SVD-169 | **0.1015** | (<0.001) | **4.25** | (0.09) | 338 | **0** |
| NNGP | 0.1036 | (<0.001) | 111.16 | 1.66 | **149.2** | **0** |
| SVGP | 0.1096 | (<0.001) | 184.19 | (2.75) | 771.44 | 98.12 |
| SVGP-CIQ | 0.1058 | (<0.001) | 72.49 | (2.13) | 481.98 | 17.68 |
| VNN | 0.1017 | (<0.001) | 23.27 | (1.39) | 952.35 | 1267.30 |
| NGD | 0.1019 | (0.001) | 57.56 | (0.29) | 342.19 | 26.83 |
| DKL | 0.1083 | (<0.001) | 22.07 | (1.17) | 691.13 | 516.39 |
| SGPR | 0.1016 | (<0.001) | 36.27 | (1.99) | 413.37 | 100.13 |
| SKI | 0.1163 | (<0.001) | 176.90 | (5.40) | 3562.44 | 32.01 |
| LOVE | 0.1090 | (0.001) | 16.67 | (0.36) | 661.32 | 515.42 |

**Simulation 5**

Finally, we display the scaling power and empirical law of our method using the same setup as Simulation 1, with a varied sample size from $20,000$ to $2,000,00$, as shown in Figure 1. Specifically, the time for HS-SVD with $n = 2,000,000$ is comparable to the time taken by SKI and LOVE when $n = 200,000$, demonstrating the strong scalability of HS-SVD to handle big data.

## 5 DISCUSSION

In this paper, we introduced the HS-SVD method for obtaining low-rank approximations of kernel matrices. We discussed the numerical advantages of this approach, particularly in reducing both computational and memory cost, while also enhancing numerical stability. As an illustrative example, we constructed the compact Matérn kernel for arbitrary dimensions, and provided insights into its derivation, highlighting its connection to the widely used Matérn Kernel. Through empirical comparisons with a variety of SOTA scalable GP methods, we demonstrated that our method significantly reduces computational time and memory requirements while achieving the best or near-best prediction MSE, in various cases.

While the HS-SVD method offers significant computational benefits, several challenges and opportunities for improvement remain. First, although our method scales exceptionally well with sample size, it does not fully overcome the curse of dimensionality. Specially, the truncation length $m$ grows exponentially with dimension $r$, which limits its efficiency in regression with high-dimensional features. One potential solution to this is integrating HS-SVD with DKL, which utilizes neural networks to learn low-dimensional feature representations of high-dimensional data, thus mitigating the curse of dimensionality. Second, our method requires the Mercer decomposition for the chosen kernel, which may not always be available. However, an approximate Mercer decomposition might be computed in such cases. Importantly, this approximation only needs to be computed once and can be reused across multiple applications, making it a manageable limitation. Third, currently implemented on a single CPU core, our method would benefit from a GPU-based implementation, allowing for further speedups. Fourth, our method can be combined with existing methods such as variational approaches for even greater scalability. Finally, beyond its application in GP, the HS-SVD method can be extended to other problems that involve using kernel matrices, such as smoothing splines and Radial Basis Function (RBF) approximations. This broader applicability opens up numerous avenues for further research, potentially expanding the impact of this method across multiple domains.

**Reproducibility statement**: In Appendix A, we provide all the material necessary to reproduce our simulations. This file includes code for generating our simulation data, training the models, and assessing their performance. Additionally, details for the proof of Theorem 3.3 are located in Appendix B.

**Ethics Statement**: Our paper does not deal with sensitive experiments, data, or any methods that can be expected to cause harm. We have no conflicts of interest and have no data privacy concerns.

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

## A   CODE AVAILABILITY

All code necessary for running simulations, generating figures, and generating tables is available as a downloadable zip file at `https://anonymous.4open.science/r/ICLR_2025_Anonymous_Submission-7011/readme`.

## B   PROOF OF THEOREM 3.3

We split Theorem 3.3 into two cases: $r = 1$ and generic $r$. We begin with the differentiability of the compact Matérn GP in the univariate case, i.e., $r = 1$.

**Theorem B.1** *When $r = 1$, the smoothness of the compact Matérn is $\beta - 2$.*

Recall that the compact Matérn kernel is defined as

$$K(x, x') = \sum_{l=1}^{\infty} 2(\alpha^2 + l^2\pi^2)^{-\beta} \sin(l\pi x) \sin(l\pi x').$$

Then, by Karhunen-Kosambi-Loeve Theorem (Stark & Woods, 1986), $f \sim \mathrm{GP}(0, K(\cdot, \cdot))$ admits the following representation:

$$f(x) = \sum_{l=1}^{\infty} Z_l \sqrt{\lambda_l} \phi_n(x) = \sum_{l=1}^{\infty} Z_l (\alpha^2 + l^2\pi^2)^{-\beta/2} \sin(l\pi x),$$

where $Z_l$s are i.i.d. standard normal random variables. Note that, for all $x_0 \in (0, 1)$,

$$\mathbb{P}\left(\lim_{l_0 \to \infty}\left[\sum_{l=l_0}^{\infty} Z_l(\alpha^2 + l^2\pi^2)^{-\beta/2} \sin(l\pi x_0)\right] = 0\right)$$

$$\geq \mathbb{P}\left(\lim_{l_0 \to \infty}\left[\sum_{l=l_0}^{\infty} |Z_l(\alpha^2 + l^2\pi^2)^{-\beta/2} \sin(l\pi x_0)|\right] = 0\right)$$

$$\geq \mathbb{P}\left(\lim_{l_0 \to \infty}\left[\sum_{l=l_0}^{\infty} |Z_l(\alpha^2 + l^2\pi^2)^{-\beta/2} \cdot 1|\right] = 0\right)$$

$$\geq \mathbb{P}\left(\lim_{l_0 \to \infty}\left[\sum_{l=l_0}^{\infty} |Z_l(l^2\pi^2)^{-\beta/2}|\right] = 0\right)$$

$$\geq \mathbb{P}\left(\lim_{l_0 \to \infty}\left[\sum_{l=l_0}^{\infty} |Z_l l^{-\beta}|\right] = 0\right) = 1 \quad \text{if} \quad \beta > 1.$$

As a result, the probability that $f(x)$ is continuous depends on the probability that the series uniformly converges on the unit interval. The series is uniformly convergent if $\beta > 1$. Taking derivatives, we get a similar uniform convergence condition:

$$\frac{d}{dx}(\alpha^2 + l^2\pi^2)^{-\beta/2} \sin(l\pi x) = \pi l(\alpha^2 + l^2\pi^2)^{-\beta/2} \cos(l\pi x) \in O(l^{-\beta+1}).$$

For uniform convergence of the first derivative, we need $-\beta + 1 < -1$, that is, $\beta > 2$. Extending the above calculation to $k$-th order derivative, we have

$$\frac{d^k}{dx^k}(\alpha^2 + l^2\pi^2)^{-\beta/2} \sin(l\pi x) = \pi^k l^k(\alpha^2 + l^2\pi^2)^{-\beta/2} = O(l^{-\beta+k}).$$

Thus, we get that $\frac{d^k}{dx^k}f(x)$ converges uniformly with probability one if $\beta > k + 1$, that is, $k \leq \beta - 2$. This is sufficient for point-wise convergence almost surely. The $L_1$ convergence is due to the following inequalities:

$$\mathbb{E}\left[|f(x)|\right] \leq \mathbb{E}\left[\sum_{l=1}^{\infty}\left|Z_l\sqrt{\lambda_l}\phi_l(x)\right|\right] \leq \mathbb{E}[|Z_1|]\sum_{l=1}^{\infty}\sqrt{\lambda_l}.$$

Now we generalize to arbitrary dimension $r$.

**Theorem B.2** *The smoothness of the compact Matérn kernel over $[0, 1]^r$ is $\beta - r - 1$.*

Recall that the compact Matérn kernel is defined as

$$K(x, x') = \sum_{l_1,..,l_r=1}^{\infty} \lambda_{l_1,..,l_r}\phi_{l_1,..,l_r}(x)\phi_{l_1,..,l_r}(x'),$$

Where $\lambda_{l_1,...,l_r} = (\alpha^2 + \pi^2(\sum_{q=1}^r l_q^2))^{-\beta}, \phi_{l_1,...,l_r}(x) = \sqrt{2}^r\prod_{q=1}^r \sin(l_q\pi x_q)$ Similary to the univariate case, by the Karhunen-Kosambi-Loeve theorem, $f(x) \sim \mathrm{GP}(0, K)$ admits the following representation:

$$f(x) = \sum_{l_1,..,l_r=1}^{\infty} \sqrt{\lambda_{l_1,..,l_r}}Z_{l_1,...,l_r}\phi_{l_1,...,l_r}(x),$$

where $Z_{l_1,...,l_r}$ are i.i.d. standard normal random variables.

We now re-enumerate as

$$f(x) = \sum_{L=1}^{\infty}\sum_{l_1+...+l_r=L} \sqrt{\lambda_{l_1,..,l_r}}\phi_{l_1,..,l_r}(x).$$

Following the same logic as B.1, to prove $C^k$ we need only prove that

$$\star := \sum_{L=1}^{\infty}\sum_{l_1+...+l_r=L} |\sqrt{\lambda_{l_1,..,l_r}}D^k\phi_{l_1,..,l_r}(x)| < \infty, \ \forall x \in [0, 1]^r.$$

First, we note that

$$\lambda_{l_1,...,l_r} = \left(\alpha^2 + \pi^2\left(l_1^2 + ... + l_r^2\right)\right)^{-\beta} \leq \left(\alpha^2 + \pi^2\left(\left(\frac{L}{r}\right)^2 + ... + \left(\frac{L}{r}\right)^2\right)\right)^{-\beta},$$

$$= \left(\alpha^2 + \pi^2\left(\frac{L^2}{r}\right)\right)^{-\beta}.$$

Thus, we have

$$\star \leq \sum_{L=1}^{\infty}\sum_{l_1+...+l_r=L}\left|\left(\alpha^2 + \pi^2\left(\frac{L^2}{r}\right)\right)^{-\beta/2}D^k\phi_{l_1,..,l_r}(x)\right|.$$

Then we note that, for the differential operator $D^k = \prod_{q=1}^r \frac{\partial^{p_q}}{\partial x_q^{p_q}}$, $p_1 + ... + p_r = k$, we have the following inequality:

$$D^k \phi \leq L^k \phi.$$

Then we have

$$\star \leq \sum_{L=1}^{\infty} \sum_{l_1 + \ldots + l_r = L} \left| \left( \alpha^2 + \pi^2 \left( \frac{L^2}{r} \right) \right)^{-\beta/2} L^k \phi_{l_1, \ldots, l_r}(x) \right|.$$

Since $|\phi_{l_1, \ldots, l_r}| \leq 1$, we then have

$$\star \leq \sum_{L=1}^{\infty} \sum_{l_1 + \ldots + l_r = L} \left| \left( \alpha^2 + \pi^2 \left( \frac{L^2}{r} \right) \right)^{-\beta/2} L^k \right| = \sum_{L=1}^{\infty} \left[ \left| \left( \alpha^2 + \pi^2 \left( \frac{L^2}{r} \right) \right)^{-\beta/2} L^k \right| \sum_{l_1 + \ldots + l_r = L} 1 \right].$$

Then by the stars and bars Theorem, we have

$$\sum_{l_1 + \ldots + l_r = L} 1 = \#\{\text{Non-negative Partitions of } L \text{ into } r \text{ parts}\} = \binom{L + r - 1}{r - 1} = O(L^{r-1})$$

Thus, with some abuse of notation,

$$\star \leq \sum_{L=1}^{\infty} \left[ \left| \left( \alpha^2 + \pi^2 \left( \frac{L^2}{r} \right) \right)^{-\beta/2} L^k \right| \binom{L + r - 1}{r - 1} \right] = \sum_{L=1}^{\infty} |O(L^{-\beta+k+r-1})|$$

To achieve absolute convergence, the condition becomes $-\beta + k + r - 1 < -1$, that is

$$k \leq \beta - r - 1.$$

## C   ADDITIONAL EXPERIMENTAL DETAILS

The simulations in this paper were performed using an Intel i9-12900H and an NVIDIA RTX 3070 TI Mobile with 16GB of RAM and 8GB of VRAM.

The training information for simulations 1 and 2 is summarized below. Note that NGD and SVGP-CIQ have two learning rates (LR).

| Method | Loss Function | Optimizer | LR | Batch Size | Epochs | Hardware |
|---|---|---|---|---|---|---|
| HS-SVD-50 | Likelihood | Nelder-Mead | NA | NA | NA | CPU |
| NNGP | Likelihood | Grid-Search | NA | NA | NA | CPU |
| SVGP | Likelihood | Adam | 0.005 | 32 | 6 | CPU & GPU |
| SVGP-CIQ | ELBO | Adam | 0.1, 0.002 | 3200 | 5 | CPU & GPU |
| VNN | Likelihood | Adam | 0.02 | 1280 | 30 | CPU & GPU |
| NGD | ELBO | Adam | 0.001, 0.1 | 3200 | 60 | CPU & GPU |
| DKL | Likelihood | Adam | 0.02 | NA | 80 | CPU & GPU |
| SGPR | Likelihood | Adam | 0.01 | NA | 250 | CPU & GPU |
| SKI | Likelihood | Adam | 0.1 | NA | 32 | CPU & GPU |
| LOVE | Likelihood | Adam | 0.1 | NA | 100 | CPU & GPU |

Table 5: Training parameters for simulations 1 and 2

The training information for Simulations 3 and 4 are summarized below. We remark that SKI was not able to train on all 80000 training points due to memory reasons, and so SKI was only trained on 40000 data in the 2 dimensional simulations.

| Method | Loss Function | Optimizer | LR | Batch Size | Epochs | Hardware |
|---|---|---|---|---|---|---|
| HS-SVD-50 | Likelihood | Nelder-Mead | NA | NA | NA | CPU |
| NNGP | Likelihood | Grid-Search | NA | NA | NA | CPU |
| SVGP | Likelihood | Adam | 0.001 | 3200 | 20 | CPU & GPU |
| SVGP-CIQ | ELBO | Adam | 0.1, 0.002 | 3200 | 10 | CPU & GPU |
| VNN | Likelihood | Adam | 0.02 | 1280 | 10 | CPU & GPU |
| NGD | ELBO | Adam | 0.01, 0.1 | 320 | 5 | CPU & GPU |
| DKL | Likelihood | Adam | 0.02 | NA | 60 | CPU & GPU |
| SGPR | Likelihood | Adam | 0.01 | NA | 350 | CPU & GPU |
| SKI | Likelihood | Adam | 0.005 | NA | 15 | CPU & GPU |
| LOVE | Likelihood | Adam | 0.1 | NA | 40 | CPU & GPU |

Table 6: Training parameters for simulations 3 and 4

