# OpenReview forum: "Scalable Gaussian Process via Hilbert-Schmidt Singular Value Decomposition"
_ICLR.cc/2025/Conference — ICLR 2025 Conference Withdrawn Submission_

### Official Review · Reviewer_a2Ye · 2024-10-15

**Soundness:** 2
**Presentation:** 3
**Contribution:** 1
**Rating:** 5
**Confidence:** 4

**Summary:**

The manuscript proposes a methodology to scale up Gaussian processes, focusing on Gaussian process regression. The core idea is to decompose the covariance matrix via Hilbert-Schmidt singular value decomposition that, in principle, can obtain a low-rank representation for free. The method is explained and a pseudocode is offered. The method is then tested in 5 different simulations.

Overall, I enjoyed reading this paper. It is well-written and easy to read. However, there are some significant shortcomings regarding novelty, the literature survey, and the test results.

**Strengths:**

- The manuscript is clearly written.
- While not novel in itself (in my opinion), the content might very well be of interest to readers.
- Scalable GPs are a hot topic.
- The method is easy to implement, so it has a good chance to be applied.

**Weaknesses:**

(A) Major:
(1) Novelty: The idea of singular value decomposition for GPs is not new and was even described in the GP cookbook (Williams and Rasmussen). The principle has been applied in various papers, including [Sivaram Ambikasaran, Daniel Foreman-Mackey, 2015] and [Drineas, P. and Mahoney, M. W. (2005)]. I fail to recognize the novelty of the proposed approach, since, in general, decomposing the kernel into eigenvalues and eigenfunctions is well-known (as is even stated in the manuscript). Please clarify what part of the "Method" section is in fact novel. One piece of potential novelty might come from the kernel which I address in the next comment.

(2) Methods for scaling Gaussian processes have recently moved away from the limitations to a particularly (often stationary) kernel design. The community has come to the revelation that flexible kernel designs are what makes GPs powerful function approximation tools.  Dictating one or the other kernel design is therefore counter-productive. Please describe how the method could be extended to non-stationary kernels.

(3) Simulation Experiments. The simulations all use one particular function (in 1d or 2d) with points on a grid or randomly distributed. At the very least the method has to be tested on some real higher-dimensional datasets. There are so many to choose from and really any will do, but one particular synthetic function is not sufficient to demonstrate performance. Also, I am concerned about the MSE error scores (MSE is not optimal but more on that later). The MSE seems very high for all methods. The function is approximately bounded by [-1,1] in 1d and [-.2,.2] in 2d; an MSE of 0.1 seems high, especially for large datasets. I might have missed something here but this was a concern. Next, assuming gridded data is pretty unrealistic and those tests don't have too much value. My suggestions for improvements are (a) new test datasets from various fields (topography, weather data, Housing data (available through scikit-learn)), robot-datasets, and so on, (b) add the CRPS and the Negative log predictive density scores to the results.

(4) Literature overview: The introduction focuses on prior work in approximate GPs. Recent work has resulted in methods to scale up exact GPs.
Exact Gaussian Processes on a Million Data Points
Ke Alexander Wang, Geoff Pleiss, Jacob R. Gardner, Stephen Tyree, Kilian Q. Weinberger, Andrew Gordon Wilson

Exact Gaussian processes for massive datasets via non-stationary sparsity-discovering kernels
MM Noack, H Krishnan, MD Risser, KG Reyes

In addition, it seems that the Vecchia approximation is neither discussed nor compared to.
Please add a short discussion to the manuscript and a reason why those methodologies were not compared.

(B) Minor:
(1) The MSE alone is not a good score to judge the performance of a GP. It should be augmented by the CRPS.
(2) Deep Kernel Learning is mixed in the intro with methods for scalability. That seems to be coming a little bit from left field, since the main purpose DKL is a flexible way to achieve non-stationarity, not scalability.
(3) The comments on ill-conditioning surprised me. With a realistic noise model and a PSD kernel, ill-conditioning should never be an issue.

**Questions:**

Suggestions:
- Describe exact approaches to scale GPs in the literature review.
- Explain better why your approach is novel, given that applying decomposition of the kernel is not.
- The Simulations should not focus on a particular synthetic function in low dimensions. It should be the MLE, RMSE, CRPS (...) of the approximation for a variety of machine-learning-relevant datasets. One of them could be a synthetic function. In the simulations, you might also compare to the Vecchia approximation (which should be presented in the intro). The Vecchia approximation is the state-of-the-art in statistics.

Questions:
- How would the method change for non-stationary kernel designs? It is my assessment that recent applied studies increasingly try to take advantage of non-stationarity.

- Are you confident about the correctness of your MSE scores? They seem large for the given function.

---

### Official Review · Reviewer_1Mfu · 2024-10-24

**Soundness:** 2
**Presentation:** 2
**Contribution:** 2
**Rating:** 3
**Confidence:** 4

**Summary:**

The paper addresses the scalability issue of Gaussian process regression (GPR), which suffers from cubic time complexity and quadratic space complexity when dealing with large datasets. The authors propose a scalable framework based on the Hilbert-Schmidt singular value decomposition (HS-SVD) to achieve a more efficient low-rank approximation of the kernel matrix, thereby reducing the time complexity to $O(nm^2)$ and space complexity to $O(nm)$. Empirical results demonstrate that the proposed framework outperforms existing methods in terms of runtime and memory usage on simulated large-scale datasets, while requiring minimal preprocessing and being accessible without GPU resources.

**Strengths:**

The main strengths of the paper are as follows:

**S1.** Proposes a method for obtaining low-rank approximations of kernel matrices based on HS-SVD, that requires minimal tuning or preprocessing and does not depend on GPU resources.

**S2.** Discusses the numerical advantages of the proposed method, particularly in reducing both computational and memory cost.

**S3.** Empirically demonstrates that the proposed method is superior to the baselines on simulated large-scale datasets.

**Weaknesses:**

There are several weaknesses of the paper:

**W1.** The novelty of the proposed method is questionable, as the HS-SVD has already been introduced and thoroughly studied in [1]. Many definitions, lemmas, and theorems in this work are derived from [1]. For instance, Definitions 2.4 and 3.4 are adapted from [1] without proper citations.

**Suggested action:** The authors should clarify their contributions in relation to [1], and ensure proper attribution for all borrowed sections, including definitions and theorems.

**W2.** The paper lacks adequate theoretical analysis and justification regarding several key aspects:
- There is no clarification on how the authors determine the truncation order (see Q1), nor is there any quantification or analysis of the corresponding trade-off (see Q2). Specifically, it is unclear how the authors chose the truncation order and whether they have tested other values to obtain the results presented in Tables 1-4.
- There is no justification for how truncation enhances stability by eliminating ill-conditioned parts of the kernel matrix (see Q6).
- The authors do not explain how the proposed method can be integrated with existing dimension reduction and variational inference methods, nor how this integration could result in faster runtime and lower memory costs (see Q8).

**Suggested action:** The authors should consider conducting an ablation study to analyze the effect of different truncation orders, and/or provide a theoretical bound on the approximation error.

**W3.** Most of the baseline methods used in experiments are relatively outdated, with many published within 5-10 years ago. The authors should consider including more recent state-of-the-art methods to evaluate their method. Some relevant works are as follows:
- Wu, K., Wenger, J., Jones, H.T., Pleiss, G. and Gardner, J., 2024, April. Large-scale Gaussian processes via alternating projection. In *International Conference on Artificial Intelligence and Statistics* (pp. 2620-2628). PMLR.
- Allison, R., Stephenson, A. and Pyzer-Knapp, E.O., 2024. Leveraging locality and robustness to achieve massively scalable Gaussian process regression. *Advances in Neural Information Processing Systems*, 36.
- Li, K., Balakirsky, M. and Mak, S., 2024, April. Trigonometric Quadrature Fourier Features for Scalable Gaussian Process Regression. In *International Conference on Artificial Intelligence and Statistics* (pp. 3484-3492). PMLR.
- Noack, M.M., Krishnan, H., Risser, M.D. and Reyes, K.G., 2023. Exact Gaussian processes for massive datasets via non-stationary sparsity-discovering kernels. *Scientific reports*, 13(1), p.3155.

**Suggested action:** The authors should consider incorporating more recent works as baselines (e.g., the ones listed above) and discuss how their method compares, both theoretically and empirically, to the updated baselines.

**W4.** The experimental validation is insufficient. While the proposed method shows superior results on simulated datasets, it lacks evaluation on real-world datasets. Additionally, most simulations only use $n = 10,000$ samples, which is not convincing enough to support the claim that their method is suitable for large-scale settings. It is recommended that the authors include experiments on larger real-world datasets to strengthen their results. Please refer to the experiments in the references mentioned against W3.

**Suggested action:** The authors should conduct additional experiments using real-world datasets, such as those from the [UCI Machine Learning Repository](https://archive.ics.uci.edu), which contain millions to tens of millions of samples, to better evaluate the scalability of their method.

**Questions:**

In addition to the weaknesses, the following questions need to be addressed:

**Q1.** In Definition 2.4, the authors state that they use a truncated form of the HS-SVD for computation. However, reference [1] suggests that truncating the series may result in deviations from the standard kernel space span. How do the authors determine the truncation order, and what measures are taken to ensure that the truncated form remains within the kernel space span?

**Q2.** Following Q1, the authors also mentioned that "the truncation strikes a balance between computational efficiency and the accuracy of the low-rank approximation". Can the authors provide a quantification or analysis of this trade-off? The authors may refer to reference [2] for a similar analysis.

**Q3.** In Section 3.1, the authors introduce the compact Matérn kernel. How does this kernel differ from the iterated Brownian bridge kernel discussed in reference [1]? In addition, [1] also mentioned that it is possible to extend their kernel to high-dimensional case by using tensor products, how does this compare to the proposed kernel in Definition 3.2?

**Q4.** In Section 3.2, the authors claimed that their proposed method can reduce the computational complexity of the matrix inversion from $O(n^3)$ to $O(m^3)$. However, this appears similar to existing low-rank approximation methods, such as the Nyström approximation, which also achieves the same complexity. Can the authors clarify how their approach is advantageous compared to these established methods? It is suggested to add a table comparing the proposed method with existing methods, in terms of the time and space complexities.

**Q5.** In Section 3.3, the authors claimed that truncating the small eigenvalues improves stability by removing ill-conditioned parts of the kernel matrix. Can the authors provide theoretical justification for this claim? The reviewer also did not find any experiments supporting this claim.

**Q6.** The authors use runtime (in seconds) and RAM/VRAM usage as metrics for performance evaluation; however, these are hardware-dependent metrics. Could the authors also provide hardware-independent metrics such as FLOPS and arithmetic intensity?

**Q7.** In Section 4, the authors claimed that "by integrating the strength of dimension reduction and variational inference methods with HS-SVD’s low-rank decomposition, we could achieve the best of both worlds—enhancing stability while maintaining fast runtime and low memory costs.” Can the authors perform an experiment/ablation study to support this claim?

### References
[1] Cavoretto, R., Fasshauer, G.E. and McCourt, M., 2015. An introduction to the Hilbert-Schmidt SVD using iterated Brownian bridge kernels. *Numerical Algorithms*, 68, pp.393-422.

[2] Griebel, M., Rieger, C. and Zwicknagl, B., 2015. Multiscale approximation and reproducing kernel Hilbert space methods. *SIAM Journal on Numerical Analysis*, 53(2), pp.852-873.

---

### Official Review · Reviewer_E6qD · 2024-10-25

**Soundness:** 2
**Presentation:** 2
**Contribution:** 2
**Rating:** 5
**Confidence:** 5

**Summary:**

This paper considers the computational issue in the GP area, which is widely known as a bottleneck in large-scale GP applications. THe paper proposes a method based on the Hilbert-Schmidt singular value decomposition that obtains a low-rank decomposition“for free”, reducing both time complexity and space complexity.

**Strengths:**

The idea is novel, using HS-SVD to help reduce computational cost.

**Weaknesses:**

1. Writing: many sections use "fast" as the title. It may lead to confusion about what the "fast" really means. You should add something related to complexity in it to make it clearer.
2. Experiment: I can not see the experiment on the real dataset. Can you explain why you do not do experiments on the real dataset?

**Questions:**

1. In your approximation of page 4, you simply do truncation of the representation. Do you have a theory for the approximation? Or small m can make the approximation not trustworthy.
2. You spend a great extent discussing the smoothness of the compact Matern kernel, is the result new?
3. Could you add some real-world datasets in the experiment parts?
4. Have you compared your method with the method in 'Sample and Computationally Efficient Stochastic Kriging in High Dimensions'? As far as I know, this work also states that they achieve the SOTA. You can include a comparison with this method in experiments, or explain why such a comparison was not included.

---

### Official Review · Reviewer_G6RK · 2024-11-05

**Soundness:** 2
**Presentation:** 2
**Contribution:** 2
**Rating:** 3
**Confidence:** 3

**Summary:**

This paper presents a new technique for Gaussian process regression based on the Hilbert-Schmidt singular value decomposition (HS-SVD) of the compact Matern kernel. The compact Matern kernel proposed in the paper is a generalization of the 1-D compact Matern proposed in Cavoretto et al., 2015 to higher dimensions. The eigenvalues and eignevectors of the compact Matern kernel can be computed easily via simple closed-form expressions and thus, it is possible to compute the Mercer decomposition of this kernel easily, unlike commonly used kernels like  Radial Basis Function (RBF), Matern, and exponential kernels.  Thus, one can find the HS-SVD of the compact Matern kernel by keeping only the first m eigenvalues and corresponding eigenvectors. Instead of computing and storing the complete $n \times n$ Kernel matrix, the storage cost is reduced to $O(nm)$. Operations like computing the inverse and log determinants of the kernel also become cheaper and the time complexity is reduced to $O(nm^2)$ from $O(n^3)$. Finally, some experiments using synthetic datasets are performed to show the effectiveness of the proposed approach.

**Strengths:**

The generalization of the 1D compact Matern kernel to higher dimensions is interesting. It will be useful to fully investigate and understand the properties of this kernel. In the experiments provided, the use of this kernel shows considerable promise. However, there are quite a few weaknesses of paper as pointed out below.

**Weaknesses:**

1) Insufficient experiments/evidence: The main value proposition of the paper seems to be the introduction of the generalized compact Matern kernel whose HS-SVD can be computed effectively. Some experiments are performed for nonlinear function approximation but much more evidence is required to accurately judge how well this method is truly outperforming. There is also no theoretical support showing how well the proposed approach is performing (see questions).

Apart from the points mentioned above, the paper doesn't clearly mention how to choose the parameter m (the dimension of the SVD) which is crucial to the method. In general, is there an empirical rule of thumb or some known theoretical bound that could guide the choice of m? The same point holds for the other parameters  of the compact Matern kernel $\rho, \alpha, \beta$.

2) Missing related work: There have been works that use SKI and take time almost linear in m where m is the number of “inducing points” and also doesn't scale exponentially with the input dimension r. See for example https://arxiv.org/pdf/2305.14451. Specifically, in the caption under Table 3, the authors note that methods like SKI cannot be sued since the the number of grid points grows exponentially with r. However, it seems that the above paper tackles this very problem.

3) The messaging of the paper seems to be confusing in some places. Most of the ideas presented in the paper like using an approximate Mercer decomposition of kernels have been widely known and studied. The main contribution of the paper seems to be the construction of the generalized compact Matern kernel whose HS-SVD can be computed effectively. But, for example, in the discussion section 5, the authors claim that they introduced the HS-SVD method and constructed the compact Matern kernel as an illustrative example. However, as pointed out above, HS-SVD (which basically follows from the Mercer decomposition) has been pretty well known. So, to claim that they introduced the HS-SVD method itself is a bit misleading I feel. Especially since they don't provide any algorithm for obtaining the HS-SVD of any other kernels.

Overall, I feel that this paper needs to be improved before it can be considered for publication at ICLR. Hence, I'm recommending a reject.

**Questions:**

Along with little experimental support, the paper also doesn't provide any theoretical support in terms of bounding the mean squared error for using the HS-SVD of the compact Matern instead of the full kernel. Can anything be said about the value of the approximate MSE due to using HS-SVD with respect to the true MSE if the full kernel were used?

---

### Official Review · Reviewer_2FbE · 2024-11-06

**Soundness:** 2
**Presentation:** 2
**Contribution:** 1
**Rating:** 3
**Confidence:** 4

**Summary:**

The paper proposes a method for Gaussian process regression (GPR) that reduces the classical problem of cubic and quadratic complexity in GPs. In particular, the work exploits the well-known Mercer decomposition of kernels, such that the covariance function can be expressed as an infinite sum of eigenfunctions. Under this condition, it is possible to use a special type of SVD to obtain an usable decomposition of the covariance matrix. Under the assumption of compact Matern kernels and the truncation of eigenvalues, the performance is compared on 4 different simulations with synthetic data.

**Strengths:**

It is always nice to see contributions to the scalability of GPs and particularly GPR. Despite the fact that I do not share some of the thoughts/ideas around the comments on the SOTA methods, I think the paper has a valuable point in certain directions. The Mercer's Theorem is well-known in the GP community and has been considered since the beginning of the interaction between GPs and ML. Perhaps, the main strength is the combination of the HS-SVD together with the pseudo-inverse taken from Pozrikidis (2014) and later the Sylvester determinant theorem. Even if there are some details or deeper analyses missing in such directions, I think that putting these methods together is a powerful message to look back to in these times when scalability matters a lot.

**Weaknesses:**

My gut feeling with the submission is that it has yet limitation and details that are not clear enough to validate the quality of the contribution. Some thoughts:

[W1] - Mercer's theorem is a well-known method, as well as the inverse approximation Sherman–Morrison–Woodbury formula which is basically a special case of the Woodbury matrix identity. The use of Sylvester's determinant is also common in the community, as far as I remember. In this regard, the HD-SVD is the only method that is kind of new to me here, but I am not yet sure if the combination of these together makes a super strong contribution.

[W2] - The criticism + comments around the whole literature of GPs are somehow vague, in the sense that so much detail and effort has been put on them, and are kind of discarded due to high-level opinions and not much precision on their issues and limitations. In this sense, GPR and its scalability is super-explored since 20y ago.. Then, what concerns me is that near-zero analysis is derived around the quality of the inverse for example, the effect of assuming the compact Matern kernel or the first paragraph in section 3.2 around the noise term and how reasonable is that. To me, this last detail is the worst one so far of the paper, in the sense that it is not super scientific (what is the order of the noise term, everyone can add to any ML model some noise to the data and observe nicer properties and it is easily modeled by an isotropic Gaussian right?)

[W3] - I do not buy the technique of truncating the eigenvalues in the HS-SVD and the justification that it stabilizes everything. How many of them are truncated? Little details on this point are added in my opinion.

[W4] - The fact that only synthetic-data simulations are used in the submission tends to be a bad sign in the ML/GP community, even more if the main motivation is scalability and large-scale data..

**Questions:**

Some questions and details that came to my mind while making the revision of the manuscript are:

Q1: Some comments are added in the review of SOTA methods and background about pre-processing and how problematic this is. What do pre-processing steps mean for the authors? and which method is it referring to?

Q2: Paragraph in L212: What are the real implications of the use of compact Matern kernel for RBFs. Are we limited on the type of covariance functions used, right?

Q3: The nugget for numerical stability… but what is actually the dimension/order of this one?

---

### Author Response · Authors · 2024-12-01

We appreciate the reviewers' detailed and helpful feedback. Addressing the required revisions will take significant effort, and due to time constraints, we have decided to withdraw our submission. Thank you again for your valuable comments, which we will incorporate as we revise our work.

---

### Note · Authors · 2024-12-01

**Comment:**

We appreciate the reviewers' detailed and helpful feedback. Addressing the required revisions will take significant effort, and due to time constraints, we have decided to withdraw our submission. Thank you again for your valuable comments, which we will incorporate as we revise our work.

**Withdrawal Confirmation:**

I have read and agree with the venue's withdrawal policy on behalf of myself and my co-authors.